Cross-sectional study of chromosomal aberrations and immunologic factors in Iraqi couples with recurrent pregnancy loss

Khamees Doaa A.
Al-Ouqaili Mushtak T. S. ph.dr.mushtak_72@uoanbar.edu.iq
Department of Microbiology, College of Medicine, University of Anbar , Al-Anbar Governorate, Ramadi , Iraq
Franco Bernardo
Electronic publication date: 2022 Feb 7
Publication date: 2022
Volume: 10
Electronic Location ID: e12801
Received 2021 Oct 25; Accepted 2021 Dec 24
Copyright: © 2022 Khamees and Al-Ouqaili
Copyright year: 2022
Copyright holder: Khamees and Al-Ouqaili
License: This is an open access article distributed under the terms of the Creative Commons Attribution License, which permits unrestricted use, distribution, reproduction and adaptation in any medium and for any purpose provided that it is properly attributed. For attribution, the original author(s), title, publication source (PeerJ) and either DOI or URL of the article must be cited.
License URL: https://creativecommons.org/licenses/by/4.0/

Keywords: Chromosomal aberrations, Immunologic factors, Recurrent pregnancy loss, Karyotyping

Funding: The authors received no funding for this work.

==============================
Background

Parental chromosomal aberrations are important causes of recurrent pregnancy loss (RPL). Some immunological factors such as antiphospholipid antibodies and interleukin-6 (IL-6) also contribute to this complication. The aim of this study was to determine the frequency of chromosomal abnormalities and to evaluate some of the immunological factors in couples with RPL from different cities in Iraq.

Methods

This study was conducted on 25 couples (50 individuals) who had more than two first trimester abortions in the past and 25 healthy females as controls. Karyotyping was performed on peripheral blood of all participants. Anticardiolipin (IgG and IgM), antiphosopholipid (IgG and IgM), lupus anticoagulant, and IL-6 were assayed. Data were analyzed using appropriate statistical tests.

Results

Chromosomal abnormalities were found in 28.0% (n = 7/25) of RPL couples. Of these five (10.0%) were female and two (4.0%) were male. The types of structural abnormalities were as follows: 45, XX, t(21; 21); 45, XX, rob (14, 15); 46, XX, add (21) (p13); 46 XY, add (21)(p13); 46, XX, 21ps+; 46, XY, per inv (9) (p11q12) and 45, XX, t(13q, 13q). No chromosomal abnormalities were found in the control group. Also, no significant differences were found in the immunological parameters of the couples with RPL and the control group.

Conclusion

In this study, karyotyping revealed a high number of chromosomal abnormalities associated with the RPL in Iraqi couples. Since identification of genetic causes of miscarriage is important for genetic counseling and educating couples about the risk of future pregnancies, it is recommended that conventional karyotyping be investigated in patients with RPL.

Introduction

It is well documented that recurrent miscarriages are failures after implantation in natural conception. They are also known as recurrent pregnancy losses or habitual abortions (Bashiri, Halper & Orvieto, 2018). Recurrent pregnancy loss (RPL) is one of the most common pregnancy complications and refers to the loss of a pregnancy before 20–24 weeks of gestation (El Hachem et al., 2017; Abbas et al., 2018). Determining the type of spontaneous abortion is a cornerstone in providing the most appropriate treatment (Alves & Rapp, 2021). Spontaneous abortion (SA) can be divided into inevitable abortion, threatened abortion, incomplete abortion, septic abortion, missed abortion, complete abortion, and recurrent spontaneous abortion (RSA) (Antonette, 2020).

Early abortion is thought to be caused by fetal chromosomal abnormalities in 50% of the study cases. The most important risk factors are advanced maternal age and women with a history of early pregnancy loss (American College of Obstetricians and Gynecologists’ Committee on Practice Bulletins-Gynecology, 2018). In Iraq, the problem of RSA has been found in a large number of women in recent years. The pattern of abnormal karyotypes and RSA has increased with the progression of environmental pollution in Iraq, which has been shown to have an impact on this situation (Fadhil & Ali, 2014; Al-Qaisi, Al-Ouqaili & Al-Hadithi, 2020). It is well documented that cytogenetic studies include a variety of causes of abortion.

It is generally accepted that the genetic factor plays a greater role in RSA. The genetic etiology of pregnancy losses is described as: (1) type of chromosomal abnormalities (2) distribution of chromosomal abnormalities compared to the number of pregnancy losses, (3) incidence of chromosomal anomalies in recurrent miscarriages, and (4) fetal aneuploidy in recurrent miscarriages (Pokale & Khadke, 2016). Most pregnancy losses are caused by fetal chromosomal abnormalities (Najafi et al., 2019). Overall, 50% to 60% of all pregnancy losses are caused by chromosomal aberrations and cytogenetic analysis is very important to make the diagnosis and detect genetic involvement in the family. Cytogenetic tests should also be recommended in couples with RSA because the result of these tests can provide important information for genetic counseling (Petrova-Tacheva et al., 2014; Najafipour et al., 2016).

On the other hand, about 15% to 20% of women suffering from recurrent miscarriage have antiphospholipid antibody syndrome (APS). There are heterogeneous types of antiphospholipid antibodies including lupus anticoagulant and anticardiolipin (IgG or IgM) which attack the phospholipids found in the membranes of endothelial cells, platelets, and other cells of the coagulation cascade. In addition to persistent antiphospholipid antibodies (APLs), pregnancy complications, thrombosis, recurrent abortions, and occasionally thrombocytopenia occur in this autoimmune disorder (Rodrigues, Soligo & Pannain, 2019). The anti-inflammatory cytokine, interleukin-6 (IL-6) has also been studied for its role in normal pregnancy and pregnancy-related complications. A constitutive level of this cytokine is detected in uterine tissues during both gestation and pregnancy. Nevertheless, its specific function in the different compartments of the uterus is still unclear (Omere et al., 2020). It is believed that IL-6 is one of the factors involved in the pathology of miscarriage. In a previous study, the levels of IL-6 in the blood serum of patients with miscarriage were significantly higher than those in normal pregnancy (Drozdzik, Szlarb & Kurzawski, 2013).

To date, there are few studies that can provide an overview of the prevalence of chromosomal abnormalities in women with RPL in Iraq, and many of them have not evaluated the immunological factors involved in this process. Therefore, the aim of this study was to investigate the chromosomal aberrations or abnormalities in Iraqi female patients with RPL using cytogenetic analysis, to determine the type of chromosomal abnormalities either structural or numerical, and to investigate the level of some immunological factors in them.

Materials and Methods

Ethical statement

This study was approved by the Medical Ethics Committee of the University of Anbar, Al-Anbar Governorate, Ramadi, Iraq (approval number 88, July, 25, 2021) in accordance with the Declaration of Helsinki. Written informed consents were obtained from all study participants.

Selection of study patients and healthy volunteers

This descriptive cross-sectional study was conducted on 25 couples (50 individuals) who had a history of more than two first trimester abortion and 25 couples (50 healthy individuals) including women without a history of abortion and their partners as a control group. Couples referred to the Department of Chromosomes and Cancer Disease Department in in Baghdad, the capital of Iraq, and to private gynecological clinics in Ramadi, western Iraq, were recruited for this study. The participants were from different cities in Iraq. Miscarriage was diagnosed based on the patient’s medical history and clinical examination. The women who had more than two miscarriages with different partners and women who had a living child, were excluded from this study. Other exclusion criteria were miscarriage related diseases such as infections, uterine abnormalities, diabetes mellitus, and hypertension. The inclusion criteria for the control group were couples with a normal pregnancy and those who had at least two healthy children. They were also not allowed to have or suffer from any familial genetic abnormalities.

Blood samples for cytogenetic analysis

Peripheral venous blood samples were collected from all study participants between October 2020 and May 2021. Blood samples (5 ml) were collected in heparinized tubes using a 5 ml syringe under aseptic conditions. The samples were cultured and evaluated for karyotyping.

Karyotyping

Chromosome culture was carried out by adding the blood sample to the artificial chromosome media (Euroclone, Pero, Italy) which were ready to use media. After 70 h of incubation at 37 °C, 100 μl of colcemid (Merck, Darmstadt, Germany) was added. After a passage of 60 min, cells were harvested (1,500 rpm for 7 min). Then, a 0.075 M KCl solution was added, mixed and incubated at 37 °C for 15 min. After centrifugation (1,500 rpm for 7 min), the hypotonic supernatant was removed. Then, 10 ml of cold fresh fixative solution (3:1 methanol: glacial acetic acid) was added dropwise to the cell pellet for the first 2 ml to the cell pellet. Centrifugation was then performed and the supernatant was removed. The last two steps were repeated until a clear pellet was obtained. Finally, the obtained cells were dropped onto clean slides and stained with Giemsa (Salih et al., 2018).

Chromosomal status was analyzed using the MetaClass Karyotyping apparatus (MICROPTIC S.L., Barcelona, Spain) and a fluorescence microscope (Euromex, Arnhem, Netherland). Metaphases and chromosome preparations from peripheral blood cultures were examined according to standard cytogenetic protocols. Cytogenetic analysis was performed by G-banding. Twenty metaphases were analyzed in all participants. In some cases, especially when abnormalities and mosaicism were suspected, the analysis was extended to 50 metaphases. Chromosomal abnormalities were reported according to the International System of Human Cytogenetic Nomenclature (McGowan-Jordan, Hastings & Moore, 2020).

Immunological assays

Immunological parameters, including anticardiolipin (IgG and IgM) and antiphosopholipid (IgG and IgM) were evaluated by enzyme-linked immunosorbent assay (ELISA) (Demeditec Diagnostics, Kiel, Germany) in all studied females according to the manufacturer’s instructions. The results were considered positive if the cut-off values of anticardiolipin IgG and IgM were more than 10 GPH-U/ml and 7 MPH-U/ml, respectively. The same values were also assumed for anti-phospholipid IgG and IgM. IL-6 was also assayed for all females by ELISA technique according to the procedure of the manufacturing kit, Elecsys IL-6, Roche Diagnostic, Germany. The normal reference range was up to 7 pg/ml. Lupus anticoagulant assay was performed according to procedure laid down by the manufacturing instructions kit, Stago, France, and the specific instrument, Genru CA45, China. The normal values for this test ranged from 31 to 41 s.

Data analysis

All variables were analyzed using SPSS software version 22 (Armonk, NY, USA). Chi-square test was used to compare the studied groups. The level of P < 0.05 was considered significant (Al-Ouqaili, Jal’oot & Badawy, 2018; Montazeri et al., 2020).

Results

In this study, the age of the 25 women with a normal delivered pregnancy ranged from 19 to 37 years with a mean ± SD: 26.36 ± 5.69 years. The age of the 25 study women with a history of RPL ranged from 17 to 43 years, with a mean ± SD: 31.68 ± 7.53 years. The distribution of study participants (patients with RPL and healthy control group) according to their age is shown in Fig. 1. The distribution of healthy and RPL women with ≤28 years was significantly different (P-value 0.0465), while it was not significantly different among participants with >28 years (P-value > 0.05). There was a direct significant correlation between the number of recurrent pregnancy losses and the age of the female partners. In other meanings, the likelihood of abortion increases with age, with a moderate positive significant correlation with a value of 0.565 and a P-value = 0.03 as shown in Fig. 2.

Figure 1 Distribution of studied participants based on their age range.

Figure 2 The significant correlation between the number of recurrent pregnancy loss and the age of the study female partners.

Cytogenetic analysis of paternal karyotyping was performed in all 100 studied participants. No chromosomal abnormalities were found in the females and their partners in the control group. However, chromosomal abnormalities were found in 28.0% (n = 7/25) of the RPL couples. Of these, 5 (10.0%) were female and 2 (4.0%) were male. All of these chromosomal abnormalities were structural and no numerical chromosomal abnormalities were detected. In one case, both the wife and her husband had structural abnormalities. The detailed abnormalities are summarized in Table 1. The types of structural abnormalities cases were as follows: 45, XX, rob(21; 21); 45, XX, rob(14; 15); 46, XX, add(21)(p13); 46, XY, add(21)(p13); 46, XX, 21ps+; 46, XY, inv(9)(p11q12); and 45, XX, rob(13; 13).

Table 1 Karyotyping findings in the couples with history of recurrent pregnancy loss.

Number of cases	Patients no.	Female karyotype	Male karyotype	Age (Year)	Number of abortion	Number of gestation	
Male	Female	
1	D3	45, XX, rob(21; 21)	46, XY	38	36	3	3	
2	D21	45, XX, rob(14; 15)	46, XY	35	33	3	3	
3	D4	46, XX, add(21)(p13)	46, XY, add(21)(p13)	21	18	3	4	
4	D8	46, XX, 21ps+	46, XY	16	17	4	4	
5	D15	46, XX	46, XY, inv(9)(p11q12)	44	43	3	4	
6	D17	45, XX, rob(13; 13)	46, XY	39	37	4	5	

The complete descriptions of all cases with the chromosomal abnormalities with a specific karyotype for each case are represented in the following (Figs. 3–9).

Figure 3 Karyotype of the female with 45, XX, t (21; 21).

This case is consistent with female karyotype with Robertsonian translocation of chromosome 21.

Figure 4 Karyotype of the female with 45, XX, rob (14, 15).

This case is consistent with female karyotype which has a Robertsonian translocation between the long arm of chromosome 14 and the long arm of chromosome 15.

Figure 5 Karyotype of the male with 46, XY, add (21)(p13).

This case is consistent with male karyotype which has an addition material attached to band 13 on the short arm of chromosome 21.

Figure 6 Karyotype of the female with 46, XX, add (21)(p13).

This case is consistent with female karyotype which has an addition material attached to band 13 on the short arm of chromosome 21.

Figure 7 Karyotype of the female with 46, XX, 21ps+.

This case is consistent with female karyotype which has an increase in the length of satellite on short arm of chromosome 21.

Figure 8 Karyotype of the male with 46, XY, per inv (9) (p11q12).

This case is consistent with male karyotype which has percentric inversion of chromosome 9 which may be a normal variant in human being.

Figure 9 Karyotype of the male with 45, XY, t (13q, 13q).

This case is consistent with male karyotype with balanced Robertsonian translocation between the long arms of both copies of chromosome 13.

Case 1 was a 36-year-old female with a history of two babies born with Down syndrome and repeated miscarriages (three abortion in the first trimester). The karyotype was 45, XX, rob (21; 21) indicating a Robertsonian translocation of chromosome 21 (Fig. 3). The karyotype of her husband (38 years) was normal (46, XY).

Case 2 was a 33-year-old female with a history of recurrent pregnancy loss (three abortion in the 1st trimester). The karyotype was 45, XX, rob (14; 15) consistent with a Robertsonian translocation between the long arm of chromosome 14 and the long arm of chromosome 15 (Fig. 4). The karyotype of her husband (35 years) was normal (46, XY).

Case 3 was a couple referred for chromosomal analysis because three pregnancies had been aborted in the first trimester. The karyotype of the male (21 years) as shown in Fig. 5 revealed the presence of additional material attached to band 21p13 as follows: 46, XY, add(21)(p13). On the other hand, the karyotype of his wife (18 years) showed the same structural changes observed in the husband: 46, XX, add (21)(p13) (Fig. 6).

Case 4 was a 17-years-old female with a history of four repeated miscarriages (four abortion in the first trimester). Her karyotype was 46, XX, 21ps+ consistent with an increase in the length of satellite on the short arm of chromosome 21 (Fig. 7). Her husband’s (28 years) karyotype was normal (46, XY).

Case 5 was 43-years-old female with a history of three recurrent miscarriages (three abortion in the first trimester). Her karyotype was normal (46, XX), while her husband had an abnormal karyotype which was 46, XY, per inv(9)(p11q12). This karyotype was a pericentric inversion of chromosome 9 which may be a normal variant in humans (Fig. 8).

Case 6 was a 37-years-old female with history of four recurrent miscarriages (four abortion in the first trimester). Her karyotype was 45, XX, rob(13; 13). This case involved a balanced Robertsonian translocation between the long arms of both copies of chromosome 13 (Fig. 9).

Immunological parameters

The distribution of antiphospholipid (IgG, IgM), anticardiolipin (IgG, IgM), lupus anticoagulant, and IL-6 levels in the female participants with RPL are shown in Table 2.

Table 2 The distribution of the values of anti-phospholipid (APL) (IgG and IgM), anti-cardiolipin (IgG and IgM), lupus anticoagulant (LA), and interleukin-6 among the females with recurrent pregnancy loss.

Patient no.	Age of female partner	Age of male partner	Karyotyping	LA normal range (31–41 s)	APL IgG normal range (≤10 GPH-U/ml)	APL IgM normal range (≤7 MPH-U/ml)	Anticardiolipin IgG normal range (≤10 GPH-U/ml)	Anticardiolipin IgM normal range (≤7 MPH-U/ml)	IL-6 normal range (≤7 pg/ml)	
D1	21	27	Normal	26.5	0.4	0.2	0.2	0.4	1.0	
D2	39	46	Normal	31.5	0.3	0.1	0.1	0.2	1.3	
D3	36	38	Abnormal	32.0	0.1	0.2	0.2	0.4	1.4	
D4	18	21	Abnormal	34.0	0.4	0.1	0.1	0.2	1.1	
D5	25	29	Normal	27.5	0.6	0.2	0.3	0.4	1.0	
D6	36	39	Normal	31.0	0.2	0.1	0.4	0.5	1.5	
D7	31	35	Normal	32.0	0.4	0.4	0.2	0.3	1.5	
D8	17	16	Abnormal	35.0	0.5	0.2	0.1	0.2	2.0	
D9	30	33	Normal	30.0	0.7	0.3	0.3	0.4	1.0	
D10	39	42	Normal	33.0	0.4	0.1	0.3	0.1	1.8	
D11	28	30	Normal	34.0	0.5	0.2	0.2	0.7	1.6	
D12	23	28	Normal	33.0	0.7	0.2	0.3	0.6	1.5	
D13	37	41	Normal	30.0	0.3	0.6	0.3	0.2	2.0	
D14	27	28	Normal	35.0	0.1	0.4	0.1	0.3	1.8	
D15	43	44	Abnormal	29.5	0.1	0.7	0.1	0.3	2.0	
D16	29	30	Normal	31.0	0.1	0.7	0.2	0.6	2.0	
D17	37	39	Abnormal	30.0	0.5	0.4	0.3	0.5	1.5	
D18	38	41	Normal	35.0	0.3	0.2	0.2	0.3	1.5	
D19	42	44	Normal	30.0	0.7	0.2	0.2	0.5	1.5	
D20	36	40	Normal	31.0	0.5	0.6	0.1	0.2	2.1	
D21	33	35	Abnormal	32.0	0.1	0.3	0.1	0.3	1.1	
D22	29	32	Normal	35.0	0.7	0.5	0.5	0.3	1.2	
D23	22	28	Normal	27.5	0.3	0.2	0.1	0.3	2.0	
D24	39	42	Normal	28.5	0.1	0.5	0.5	0.5	2.0	
D25	37	39	Normal	30.0	0.3	0.3	0.1	0.2	2.2	

The aforesaid values in the RPL patients and females in the control group were all within the normal ranges and no test was abnormal. As shown in Table 3, no statistical differences were found in the immunological factors of the study between the PRL patients and the healthy female subjects (P-value > 0.05).

Table 3 Comparison of anticardiolipin (IgG and IgM), antiphosopholipid (IgG and IgM), lupus anticoagulant, and IL-6 levels in the females with RPL and healthy females in control group.

parameters	Study cases/Control	Number	Mean	Std. deviation	P-value Sig. (2-tailed)	
Lupus anticoagulant	Case	25	31.360	2.4855	0.287	
Control	25	30.688	1.8833	
Antiphospholipid (IgG)	Case	25	0.372	0.2092	0.395	
Control	25	0.324	0.1855	
Antiphospholipid (IgM)
.	Case	25	0.316	0.1886	0.083	
Control	25	0.234	0.1365	
Anticardiolipin (IgG)	Case	25	0.220	0.1225	0.639	
Control	25	0.204	0.1172	
Anticardiolipin (IgM)	Case	25	0.356	0.1530	0.086	
Control	25	0.284	0.1375	
Interleukin-6	Case	25	1.584	0.3859	0.30	
Control	25	1.372	0.2762	

The results of RPL females revealed that the mean ± standard deviation of anti-phospholipid (IgG and IgM), anti-cardiolipin (IgG and IgM), lupus anticoagulant, and IL-6 were 0.37 ± 0.209, 0.36 ± 0.188, 0.22 ± 0.122, 0.35 ± 0.15, 31.36 ± 2.48, and 1.58 ± 0.38, respectively (Table 4 and Fig. 10).

Figure 10 The mean ± standard deviation of lupus anticoagulant, antiphospholipid IgG and IgM, anticardiolipin IgG and IgM, and interleukin-6 in the study women with recurrent pregnancy loss.

Table 4 The mean ± standard deviation of antiphospholipid IgG and IgM, anticardiolipin IgG and IgM, lupus anticoagulant, and interleukin-6 among the females with recurrent pregnancy loss.

Item	No.	Minimum values	Maximum values	Mean	Standard deviation	
Age	25	18	44.0	31.92	7.251	
Antiphospholipid IgG	25	0.1	0.7	0.37	0.209	
Antiphospholipid IgM	25	0.1	0.7	0.32	0.189	
Anticardiolipin IgG	25	0.1	0.5	0.22	0.123	
Anticardiolipin IgM	25	0.1	0.7	0.36	0.153	
Lupus anticoagulant	25	26.5	35.0	31.36	2.486	
Interleukin-6	25	1.0	2.2	1.58	0.386	

Discussion

It is well documented that recurrent pregnancy loss is a multifactorial condition with many causes including parental genetic problem, uterine abnormalities, immunologic disorders, hematologic disorders, hormonal imbalances, and environmental factors. Spontaneous miscarriage in the first trimester is mainly due to chromosomal anomalies, such as single gene mutations, chromosomal instability and sperm chromosomal abnormalities, which explain idiopathic reproductive loss (Hardy & Hardy, 2018; Agenor & Bhattacharya, 2015).

Several studies have been carried out to determine the prevalence of chromosomal aberrations among women with spontaneous miscarriage. However, this issue is rarely addressed in Iraqi couples with RPL. In this study, chromosomal abnormalities were detected in 24.0% (n = 6/25) of Iraqi couples with RPL. Moreover, chromosomal abnormalities were higher in females (10.0%) than in males (4.0%). In a previous study by Hanif et al. (2019) from Pakistan, 28.1% of the studied couples with recurrent spontaneous miscarriages had chromosome abnormalities which was nearly eqal to this study. They also reported chromosomal abnormalities in 55.6% of females and 44.4% of males, indicating almost equal prevalence in both genders, which was in contrast to the current research.

In a review article by Marqui (2018), the percentage of chromosomal abnormalities in couples with recurrent miscarriages varied from 1.23% to 12.0%. He claimed that in almost 70.0% of the studies, chromosomal abnormalities were above 50%. In a study from Iran by Narooie-Nejad et al. (2017), chromosomal abnormalities were detected in 6.7% of 30 couples with a history of recurrent miscarriage, which was lower than our findings. They detected the chromosomal abnormalities in two females, while no males had these abnormalities, which was contrary to the current study. In another research from Iran by Ghazaey et al. (2015), 11.7% of 728 couples with a history of recurrent miscarriage had abnormal karyotypes. Moreover, the number of females (n = 48) with chromosomal aberrations was higher than that of males (n = 37), which was consistent with the current study. The prevalence of chromosomal aberrations in the current study was higher than the research by the Fan et al. (2016) from China. They reported a prevalence rate of 2.98% for chromosomal abnormalities in 1,948 couples with RSA. However, their study showed a higher incidence rate of chromosomal abnormalities in females than in males, which was in agreement with our findings. Also, Elkarhat et al. (2019) from Morocco reported a lower prevalence rate of chromosomal abnormalities (11.0%) in 627 couples with RSA compared to the current study. Nevertheless, in line with our findings, the incidence rate of chromosomal aberrations was higher in women (7.5%) than men (3.5%) (Elkarhat et al., 2019). The single ovum produced each month is a possible reason for the higher chromosomal abnormality in the female partner. Nature then selects against the abnormal gametes, as millions of sperm are released with each expulsion (Fan et al., 2016).

Despite the clear definition of RPL as the natural termination of two or more consecutive pregnancies before the fetus is capable of external life (20th week), our study was designed to include more than two consecutive pregnancy losses and exclude those with a history of two recurrent abortions to increase the selectivity of cases of chromosomal abnormalities with high suspicion. Thus, the relatively high prevalence chromosomal abnormalities was found in 20.0% (5/25) of female partners and 8.0% (2/25) of male partners, compared to other studies that found chromosomal abnormalities to be common in approximately 2% to 8% of couples with RSA (Fan et al., 2016).

In this study, structural chromosomal abnormalities were detected in seven patients with RPL. The presence of structural chromosomal abnormalities and lack of numerical abnormalities were other findings in this study that were not predictable. The lack of numerical abnormalities in this study was in contrast to the previous studies by Hanif et al. (2019) from Pakistan, Ghazaey et al. (2015) from Iran, and Elkarhat et al. (2019) from Morocco. The structural chromosomal abnormalities encountered in this study were as follows: balanced Robertsonian translocation (2/7), unbalanced Robertsonian translocation (1/7), pericentric inversion (1/7), polymorphic variants (1/7), and addition (2/7). This study showed that Robertsonian translocation was the most prevalent chromosomal abnormality in the couples with RPL. In contrast to our findings, several studies showed that reciprocal chromosomal translocations were the most common chromosomal alterations in couples with RPL (Ghazaey et al., 2015; Fan et al., 2016; Elkarhat et al., 2019). Although reciprocal chromosomal translocations were not present in this study, similar to several previous studies, the Robertsonian translocation was identified as one of the predominant aberrations in RPL cases (Ghazaey et al., 2015; Elkarhat et al., 2019). Our study indicated a prevalence rate of 12.0% (n = 3/25 couples) for Robertsonian translocation among patients with RPL, which was lower than previous studies from Iran (9.4%) (Ghazaey et al., 2015) and Morocco (1.43%) (Elkarhat et al., 2019). It is estimated that 2% to 5% of couples with recurrent miscarriages carry a balanced chromosomal abnormality, which most commonly manifests as a balanced reciprocal rearrangement or Robertsonian translocation (Priya et al., 2018). In a study by Priya et al. (2018) from India, among 76 couples with RPL cases, 3.2% (n = 5/152 individuals) had abnormal chromosomes, of which 4 (80%) had balanced reciprocal translocations and none had Robertsonian translocations. These findings were in contrast to the results of the current research. In the present study, three cases of Robertsonian translocations were detected, affecting chromosomes 13, 14, and 21. About 75.0% of all Robertsonian translocations affect chromosomes 13 and 14 that are the most prevalent forms (Fan et al., 2016).

Another chromosomal abnormality found in 2% (n = 1/50) of study participants with RPL involved pericentric inversions in chromosome 9. Several studies have shown an association between inversion 9 and infertility, recurrent abortions, and abnormal phenotypes (Ghazaey et al., 2015; Elkarhat et al., 2019; Abdi et al., 2018). However, the prevalence rate of inversion in this study was lower than previous studies from Iran (31.8%) (Ghazaey et al., 2015) and Morocco (4.3%) (Elkarhat et al., 2019).

In this study, statistical analysis revealed that there was a direct significant correlation between the number of RPL and the age of the female partners, and the likelihood of abortion increased with age. This finding was in contrast to a previous study from Pakistan (Hanif et al., 2019) but in line with a study from China (Sheng et al., 2021). Further, scientists believe that marriage and pregnancy of a mother at older age increase the risk of abortion, fetal and chromosomal problems, and pregnancy-related complications. Therefore, it is recommended that pregnant women of older age should undergo regular checkups and tests on natural fetal development (Moradinazar et al., 2020). Like our study, Jisha, Sanuj & Dinesh Roy (2014), also concluded in their study that the couples who reported for RPL had higher percentage of abnormal karyotypes. The abnormal karyotype was higher in couples with older age, longer duration of marriage, higher number of pregnancies, and higher number of spontaneous abortions.

To the best of our knowledge, there are few studies that have investigated the level of immunologic factors in the Iraqi RPL population. A strength of the current study was to evaluate the levels of various immunological factors to find an association between RPL and these markers in Iraqi patients. The results revealed that all immunological factors including antiphospholipid (IgG, IgM), anticardiolipin (IgG, IgM), lupus anticoagulant, and IL-6 were within the normal range in both RPL and control groups and no test was abnormal. There were no significant differences between two groups, suggesting an association between elevated immunological factors and RPL cases. In contrast to these findings, a previous study by Jabber, Hassan & Abdullah (2020) from Iraq, showed a highly significant increase in antiphospholipid, anticardiolipin, and lupus anticoagulant concentration in females with recurrent miscarriages compared to the control group. Also, Drozdzik, Szlarb & Kurzawski (2013), reported that patients with spontaneous miscarriage had significantly higher levels of IL-6 in their blood serum than those with physiological pregnancy. Similarly, Al-Sherbeny & Hassaan (2017) from Egypt showed a significantly higher serum level of IL-6 in women with RPL than in the control group. In another study by Kniotek et al. (2021) from Poland, RPL patients had significantly lower levels of the proinflammatory cytokine IL-6 compared to healthy women. In another study from Iraq, in line with our study, no significant differences in the IL-6 levels were found between women with miscarriage and control groups (Abdullah & Mahdi, 2013).

Conclusions

This study revealed a high prevalence of chromosomal abnormalities in Iraqi couples with RPL compared to other countries. Robertsonian translocation was the most common chromosomal abnormality. All chromosomal abnormalities were structural and no numerical aberrations were detected. Chromosomal analysis is a necessary component of etiologic research in couples with recurrent miscarriages. Since identifying genetic causes of miscarriages is important for genetic counseling and educating couples about the risk of future pregnancies, it is recommended that conventional karyotyping be evaluated in patients with RPL in Iraq. Finally, no significant differences in the levels of antiphospholipid, anticardiolipin, lupus anticoagulant, and IL-6 were found between RPL patients and the control group in this study. One of the limitations of this study was the small sample size due to traffic restrictions caused by the coronavirus disease 2019 (COVID-19) pandemic. Therefore, it is recommended to perform this study in more regions and with a larger sample size of the country in order to confirm or reject the obtained results. Another limitation was the lack of assay for other immunological factors including natural killer cells (NK-cells), regulatory T cell (Treg), dendritic cells, plasma cells, and human leukocyte antigen.

Supplemental Information

Supplemental Information 1 Raw data: Female patients.

Click here for additional data file.

Supplemental Information 2 Raw data: Age groups.

Click here for additional data file.

Supplemental Information 3 Il-6 raw data.

Click here for additional data file.

Supplemental Information 4 Raw data: Immunological factors.

Click here for additional data file.

Supplemental Information 5 Il-6 raw data.

Click here for additional data file.

Supplemental Information 6 Supplemental File.

female raw data

Click here for additional data file.

Additional Information and Declarations

Competing Interests

Author Contributions

Human Ethics

Data Availability

The authors declare that they have no competing interests.

Doaa A. Khamees conceived and designed the experiments, performed the experiments, analyzed the data, prepared figures and/or tables, authored or reviewed drafts of the paper, and approved the final draft.

Mushtak T. S. Al-Ouqaili conceived and designed the experiments, performed the experiments, analyzed the data, prepared figures and/or tables, authored or reviewed drafts of the paper, and approved the final draft.

The following information was supplied relating to ethical approvals (i.e., approving body and any reference numbers):

This study was approved by the Medical Ethics Committee of the University of Anbar, Al-Anbar Governorate, Ramadi, Iraq (approval number 88, July, 25, 2021) following the Declaration of Helsinki. The written informed consents were obtained from all study participants.

The following information was supplied regarding data availability:

Raw data are available as Supplemental Files.

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
