# Peer review of "Cross-sectional study of chromosomal aberrations and immunologic factors in Iraqi couples with recurrent pregnancy loss"

_PeerJ, doi:10.7717/peerj.12801_

## Round 0.1 · original submission · Major Revisions

Dear authors:
We have received the reports of two experts in the field, and they have identified numerous aspects that need revisions. The urgent matters are the statistical analysis to clearly show differences between the two groups on all immunological factors. Also, I kindly request that you justify the sample size adequately. Finally, please address all the concerns raised by the reviewers point-by-point.

Thank you for submitting your work to Peer J.

Best regards

·

Basic reporting

Please see my additional comments.

Experimental design

Please see my additional comments.

Validity of the findings

Please see my additional comments.

Additional comments

The manuscript entitled “Cross-sectional study of chromosomal aberrations and immunologic factors in Iraqi couples with recurrent pregnancy loss” by Doaa A. Khamees et al. investigated the chromosome abnormalities and immunological factors using peripheral blood from 25 couples with more than two RPL in the first trimester from Iraq in comparison to 25 healthy females. They performed Karyotyping to detect structural abnormalities and ELISA to measure Anticardiolipin (IgG and IgM), antiphosopholipid (IgG and IgM), lupus anticoagulant, and IL-6. They have found that chromosomal abnormalities were prevalent in 28.0% of RPL couples, while control females had no chromosomal abnormalities. Immunological factors were not significantly differential between RPL and control group. The study is of potential significance and emphasizes the importance to conduct karyotyping in patients with RPL.
1. In Text Line 102, “This cross-sectional descriptive study was done on 25 couples (50 individuals) who had a history of more than two abortion during the first trimester and 25 healthy females without a history of abortion as the control group”. What are the inclusion criteria of control women?
2. Table 1 was not described and interpreted in the manuscript. It showed that distribution of healthy and RPL women was similar in different age ranges (P-value>0.05 in >28 years). Authors could present the data using stacked bars, which is straightforward.
3. Figure 1 demonstrated that the number of abortion was increased with the age progression at which a moderate positive significant correlation was seen”. Is it just because older women tried more pregnancies (thus more miscarriage) than younger women?
4. In text line 170, “Case 1 was a 36 years old female with history of two babies born with Down syndrome and repeated miscarriage”. It should be “36-year-old female”.
5. How will the aberrant karyotypings in the manuscript guide clinical decision? If patients are diagnosed to bear aberrant chromosomes that induce RPL, what is the clinical suggestion?
6. Did authors measure anti-phospholipid (IgG and IgM), anti-cardiolipin (IgG and IgM), lupus anticoagulant, and IL-6 in healthy women? Are the levels in RPL women significantly higher than healthy women?
7. In addition to immunological factors above, other diagnostic parameters were also reported to differ between RPL patients and healthy women, including NK-cells (PMID: 24285824, PMID: 21613313), Treg (PMID: 19766059, PMID: 20331584), dendritic cells, plasma cells, and human leukocyte antigen (PMID: 18263639). Are they differentially expressed in RPL patients compared to control group in the manuscript?

Reviewer 2 ·

Basic reporting

There are many grammatical mistakes throughout the manuscript, the authors should proofread and seek help form professional editors.

Experimental design

1. The authors need to increase the sample size to reach solid conclusion.

2. The authors should chose a comparable number of participants in each age range in the studied and control group. The number of the participates younger than 28 and older than 34 are quite different between the two groups.

3. Instead of using 25 women with no history of abortion as control group, the authors should also include and do karyotype analysis on their male partners.

Validity of the findings

1. The authors should deleted the 5 cases of female with abnormal karyotypes from the correlation analysis in figure 1 since the recurrent pregnancy loss in these women is mainly caused by their chromosomal mutations but not age.

2. This reviewer is confused by "All of the above values were lower than those observed for the cut-off values and considered negative for these parameters" in line 202-203 and "The results revealed......were in normal ranges in both RPL and control groups" in line 286-288 and thought these conclusions are in conflict with each other. The authors should provide the normal range for all immunological factors measured in the study as part of table 3.

3. The authors can not just conclude no differences between the two groups on all immunological factors without showing the statistical analysis results to the readers.

4. In line 217, the should be 24% (n=6/25) in this study.

5. In line 218, the chromosomal abnormalities should be 20% (5/25) in female and 8% (2/25) in males.

6. In line 263, the prevalence rate should be 12%

Additional comments

The authors should summarize the findings from current available studies on RPL in Iraq couples and/or those from other regions and include a table that lists all important factors such as age range, number of abortion, karyotype abnormality rate, etc to help readers better understand and appreciate the value of this study. In the current discussion, the authors were merely stating that their numbers are lower or higher than other studies without any insightful analysis on the possible reasons for these differences. Since the authors purposely chose women with more than 3 abortion, their chromosomal abnormality could be higher than those with less than 3 abortions in other studies, but without providing these information, the readers wouldn't know. These also applies to the immunological factor analysis, a table that summarized those parameters in available studies should make this one more valuable and meaningful.

---

## Round 0.2 · accepted · Accept

Dear authors,

Thank you so much for providing a revised manuscript addressing the major concerns of the reviewers. After careful revision, me and the reviewer find the manuscript suitable for publications. I thank you for choosing PeerJ for your paper.

Best regards

·

Basic reporting

Authors have successfully responded to all my comments.

Experimental design

None.

Validity of the findings

None.